# Coupled Electric and Hydraulic Control of a PRS Turbine in a Real Transport Water Network

**Marco Sinagra [1],\*** , **Costanza Aricò [1]** , **Tullio Tucciarelli [1]** , **Pietro Amato [2]**
**and Michele Fiorino [3]**

1   Dipartimento di Ingegneria, Università degli Studi di Palermo, viale delle Scienze, 90128 Palermo, Italy;
    costanza.arico@unipa.it (C.A.); tullio.tucciarelli@unipa.it (T.T.)
2   WECONS (Water Engineering CONSulting) company, via Agrigento n.67, 90141 Palermo, Italy;
    p.amato@wecons.it
3   Layer Electronics company, S.P. KM 5,3 C.da S. Cusumano, 91100 Erice (TP), Italy; info@layer.it
*   Correspondence: marco.sinagra@unipa.it; Tel.: +39-091-238-96518

**Abstract:** Although many devices have recently been proposed for pressure regulation and energy harvesting in water distribution and transport networks, very few applications are still documented in the scientific literature. A new in-line Banki turbine with positive outflow pressure and a mobile regulating flap, named Power Recovery System (PRS), was installed and tested in a real water transport network for the regulation of pressure and flow rate. The PRS turbine was directly connected to a 55 kW asynchronous generator with variable rotational velocity, and coupled to an inverter. The start-up tests showed how automatic adjustment of the flap position and the runner velocity variation are able to change the characteristic curve of the PRS according to the flow delivered by the water manager or to the pressure set-point assigned downstream or upstream of the system, maintaining good efficiency values in hydropower production.

**Keywords:** pressure control; micro-hydropower; energy recovery; water distribution network; banki turbine; energy harvesting

## 1. Introduction

Although many cities continue to use fossil fuels as their main energy source, the use of renewable energy sources [1] is becoming a key political solution to mitigate climate changes occurring in the world. In this context the economic and social value of water is due today not only to its domestic and agricultural use, but also to the potential energy embedded in its delivery to low-altitude urban areas [2,3]. Water distribution or transport networks have been traditionally designed to meet consumer demands, usually variable over time, at the outlet of the pipe network, while keeping the pressure within a given pressure range, to provide a high-quality service level. Recently, new design approaches have also been based on additional hydraulic parameters such as resilience [4]. In both cases, to control flow rate and pressure in the water network, water managers very often install pressure reducing valves (PRV) and needle valves along the pipelines. PRVs are aimed to control pressure in the conduit for a given demand and needle valves are used to control flow rate given fixed outlet pressure [5–8]. An alternative to the use of valves is the use of pumps as turbines (PATs), or small hydraulic turbines [9] to convert hydraulic energy into electricity as an alternative to dissipation.

Nowadays many studies can be found in the literature about the use of turbines with free outlet discharge [10–14] or positive outlet pressure [15]. However, the use of these turbines is limited by their high cost compared to the gross power usually available in the pipelines. For these applications, less expensive solutions include crossflow mini-turbines [14] in the case of free outlet discharge and

PATs [16,17] in the case of positive outlet pressure. The main drawback of PATs is given by the need to dissipate part of the available energy when the flow rate or head jump values required by the water manager are different from the design ones, due to the absence of any hydraulic system to control the characteristic curve [18]. In order to maintain hydraulic control of the network, PATs [19,20] and crossflows [21] are often coupled with electronic systems for regulation of runner rotation velocity or with installation of PRV valves in series or parallel with the PAT [22]. This type of solution to produce energy from hydro sources is also applied for the recharge of electric vehicles in urban areas [23].

An alternative, more efficient and also less expensive way to produce energy from hydro sources while keeping the hydraulic control of the network is given by a new crossflow type of turbine, named PRS and already proposed by the authors in previous numerical [24] and laboratory experimental studies [25]. PRS has the simplicity of crossflow turbines but is also equipped with a hydraulic regulation system which allows changes in the characteristic curve according to the specific flow rate or to the head jump required by the water manager.

The most relevant difference between micro-turbines like PRS and PATs is that the PRS turbine attains the set-point required by the water manager by simply regulating the flap position. The same result requires, for PAT, to dissipate energy and/or to bypass a part of the flow rate, or to change the rotational velocity with a strong reduction of its efficiency.

In this paper, the design and installation of a 55 kW PRS turbine in a Sicilian aqueduct, and the start-up tests subject to flow rate and pressure variations, are described and analyzed for the first time.

## 2. PRS Turbine

The PRS turbine is a new in-line crossflow type of micro-turbine, with positive outflow pressure and a mobile regulation flap for the hydraulic control of the characteristic curve, developed and tested by the authors at the hydraulic laboratory of the University of Palermo [24–26].

A PRS turbine has five main components (Figure 1), namely: the convergent pipe, the nozzle, the mobile flap, the rotating runner and the pressurized diffuser. The convergent pipe is aimed to accelerate the particles, transforming most of the potential pressure energy into kinetic energy, and the nozzle works as a/the distributor of the flow rate entering the runner through the inlet surface. The mobile flap varies the inlet surface in the runner in order to control the velocity of the inlet particle during any change in the flow rate and to keep constant the ratio between the tangent velocity component of the particle and the runner rotational velocity at the same inlet location. The runner inlet and outlet surfaces are part of a cylinder, with generator lines parallel to the axis and laterally bounded by the two runner disks. The two runner disks form a single solid block with the blades, which are semi-circular and have a constant inner radius. Water flow goes through the blade channels twice, before leaving the runner and entering the diffuser section. This part, which is missing in the original crossflow turbine for zero-pressure outlet flow, is designed in order to minimize dissipation of the particle-specific energy along the path between the runner and the outlet section of the turbine case. The PRS turbine can be set in the "passive" or "active" mode. In the "passive" mode the device is used to set the piezometric level at any required value, lower than the inlet one, but even much greater than the ground elevation, while also being variable in time. In the "active" mode, the device is used to set the flow rate at any required value by controlling the flap position and the pressure reduction occurring between the inlet and outlet pipe sections.

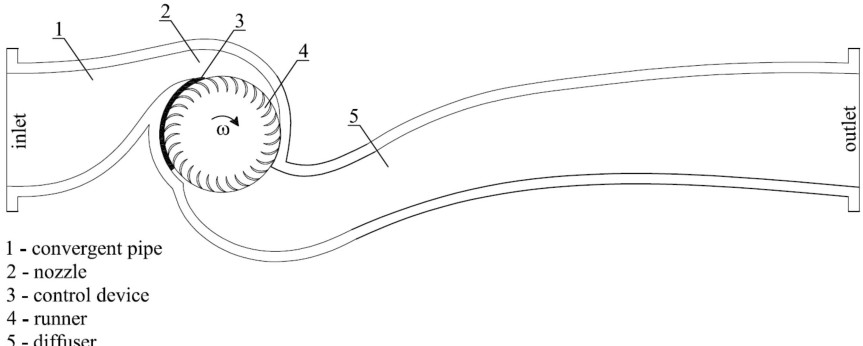

1 - convergent pipe
2 - nozzle
3 - control device
4 - runner
5 - diffuser

**Figure 1.** Vertical section of a PRS turbine.

Turbine design has to satisfy three conditions assigned at the Best Efficiency Point (BEP) among the runner diameter $D$, the rotational velocity $\omega$, the flow rate $Q$ and the net head $\Delta H$ occurring between the inlet and the outlet pipes. The first equation is the energy conservation equation, which according to previous studies ([24–26]) is given by:

$$V = C_V \sqrt{2g\left(\Delta H - \xi \frac{\omega^2 D^2}{8g}\right)} \qquad (1)$$

where $V$ is the velocity norm at the runner inlet surface, $C_V = 0.98$, $\xi = 2.1$ and $g$ is the gravitational acceleration.

The second equation is the mass conservation equation, which provides:

$$Q = \frac{BD\lambda_{rmax}V \sin\alpha}{2} \qquad (2)$$

where $B$ is the runner width, $\lambda_{rmax}$ is the maximum inlet angle, equal to 110°, and $\alpha$ is the angle between the particle velocity and the tangent direction at the runner inlet (Figure 2), approximately equal to 15°. The third equation is the optimality condition of the velocity ratio $V_r$, defined as the ratio between the tangent component of the inlet velocity and the runner rotational velocity at the same inlet surface, that is:

$$V_r = \frac{DV \cos\alpha}{2\omega} \qquad (3)$$

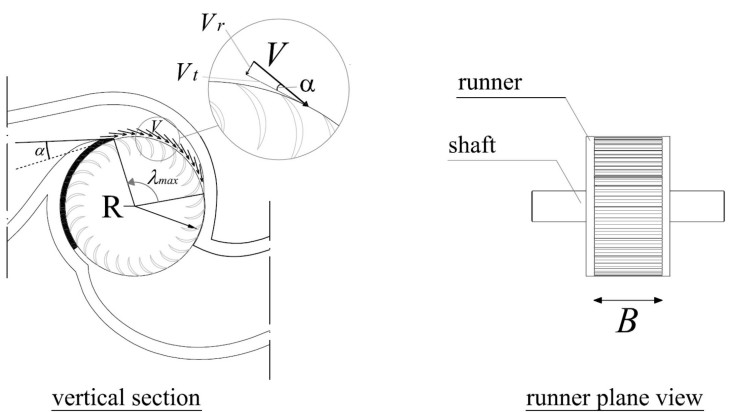

vertical section          runner plane view

**Figure 2.** Nozzle and runner geometry of PRS turbine.

Sinagra et al. [24] showed that the maximum efficiency in PRS turbine is obtained assuming $V_r = 1.7$.

The diameter $D$ and width $B$ can be found by fixing in Equations (1) and (3) the rotational velocity $\omega$, and by solving the system of Equations (1)–(3) in the unknowns $V$, $D$ and $B$. This is the commonest approach for the design of mini-hydro turbines, where the runner is directly connected to the shaft of the asynchronous electric generator, which has a fixed rotational velocity.

## 3. Electrical Energy Production and Velocity Regulation

In small-scale hydroelectric plants, with power lower than 250 kW, the simplest way to convert hydraulic power into electrical power is to couple an asynchronous three-phase generator to the turbine runner. In case (A), when the electric generator is directly connected to the AC grid, the reactive power required by the electrical generator to properly operate is provided by the grid itself, while in case (B), that of a stand-alone plant, the reactive power is provided by a local capacitor bank. The choice of the asynchronous generator is motivated by its simplicity and robustness. However, in both operation modes A and B, the rotational velocity of the electric generator is closely related to the frequency $f$ of the AC grid (grid-connected) or of the electrical equipment (stand-alone), which in Europe is equal to 50 Hz, through the equation:

$$\omega = \frac{60 f}{2p} \tag{4}$$

where $\omega$ is the rotational velocity in rotations per minute and $p$ is the number of poles.

When the net head $\Delta H$ changes along with the operating conditions of the hydraulic network, Equations (1) and (3) cannot be satisfied together with same diameter $D$, unless the runner rotational velocity $\omega$ is changed. For this reason, the rotational velocity of the runner is optimized by means of an electric system. The electric regulation system consists of a rectifier and an inverter. The task of the rectifier is to convert the alternating voltage supplied by the asynchronous three-phase generator, working at variable voltage and frequency, into a continuous voltage for the inverter power supply. The inverter adopted is a total-control Insulated Gate Bipolar Transistor (IGBT) bridge in configuration B6 (three branches in parallel, each one with two IGBTs in series), which commutes the continuous voltage supplied by the rectifier into a sinusoidal alternating voltage at 50 Hz. The reactive power required by the electrical generator is provided in the stand-alone case by a local capacitor banks cabinet with automatic power control (Figure 3).

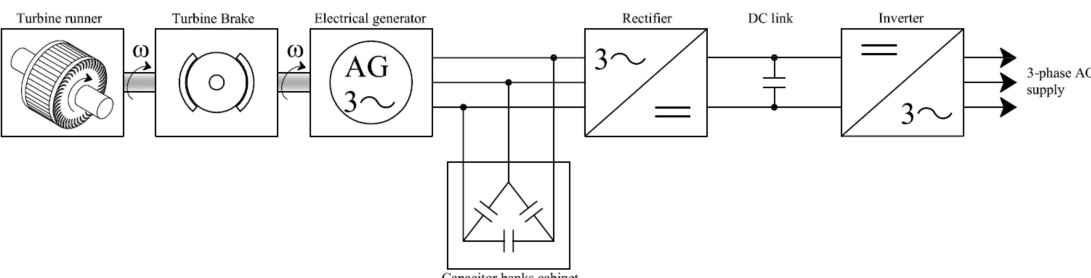

**Figure 3.** Block diagram of a direct drive power conversion unit.

With this configuration, the optimal rotational velocity $\omega$ of the runner is automatically attained in case B by regulating the voltage coming out of the inverter. Higher electric loads will lead to higher power, but also to a reduction of the turbine rotational velocity, due to a torque resistance increment. This implies that the system will shortly reach an equilibrium condition that will change, along with the power delivered in the network, as a function of the given voltage.

A similar scheme can be attained in case A, by disconnecting the capacitor banks cabinet and regulating the current coming out of the inverter.

## 4. Study Case: Gela-Aragona Aqueduct

We investigated the design and management of a PRS turbine inline of an oversized water transport network, subject to continuous flow rate regulations due to the changing demand of water users.

The water transport network, called the Gela-Aragona aqueduct, is part of the larger Water Transport Network of Sicily (Italy). The Gela-Ragona aqueduct starts from an upper tank, called "Belvedere" and located at an altitude of 460 m above sea level, supplying a lower tank named "Forche", located 335 m above sea level. This tank supplies the water distribution network of the city of Agrigento, as well as another tank located at an altitude of 75 m above sea level, serving the water distribution network of the town of Licata. Along the pipeline there are two pressure maneuvering buildings, called "Fontes Episcopi" and "San Biagio Mendolito", and between them there is a derivation supplying a small urban center (Figure 4). The flow rate from the "Belvedere" reservoir changes in the range 70–100 L/s and is regulated at present by a needle valve located immediately downstream of the reservoir. Inside the mentioned flow rate range the pressure measured at the "Fontes Episcopi" building changes in the range 0.2–0.6 MPa. If the pressure measured at "Fontes Episcopi" is above 0.5 MPa, the "Forche" tank is filled; otherwise the flow is conveyed entirely to the Licata tank.

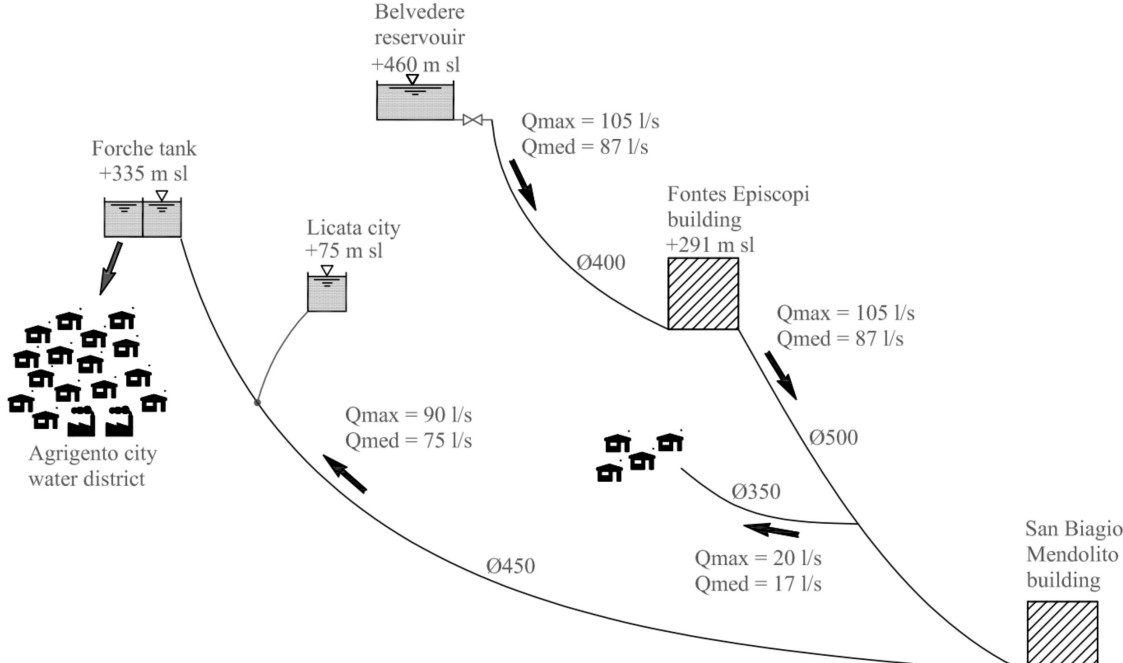

**Figure 4.** Scheme of the water transport network.

Inside the mentioned flow rate range, the pipeline connecting the "Belvedere" reservoir to the "Fontes Episcopi" building, which is 3.5 km long, is not completely full and the pressure drop $\Delta H$ of the free surface transition section inside the pipeline, with respect to the piezometric level at the "Fontes Episcopi" building, is approximately proportional to the square of the flow rate released through the needle valve by the water manager (Figure 5).

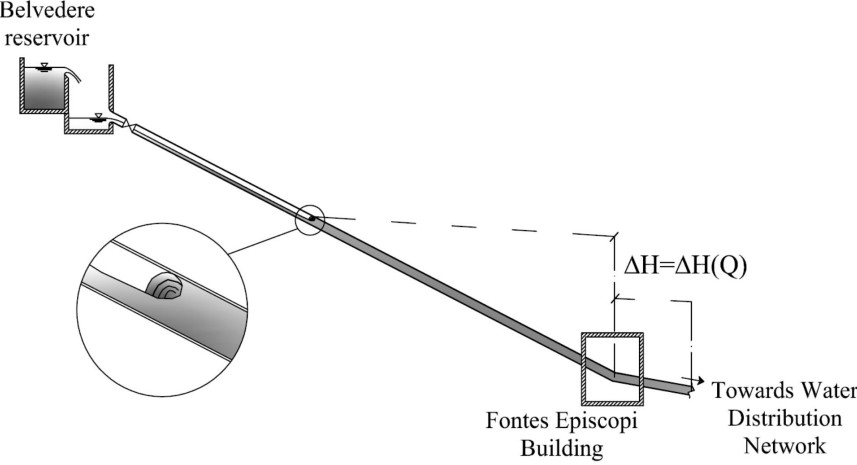

**Figure 5.** Hydraulic regime inside the upstream pipeline without the PRS turbine.

These operating conditions provide a hydraulic jump available for hydroelectric production between the surface transition and the "Belvedere" reservoir, which can be converted into electricity by a PRS turbine installed inside the Fontes Episcopi building at an altitude of 291 m above sea level. The maximum electricity production would occur in the case of a fully pressurized pipe, with head losses equal to 9.00 m in the case of a maximum flow rate. In order to guarantee the maximum flow rate when the maximum pressure occurs at Fontes Episcopi (0.6 MPa = 60m), the following values were assumed in Equations (1)–(3) for the design of parameters $D$ and $B$ in the condition of a fully opened flap: $\Delta H = 100$ m and $Q = 105$ l/s.

Assuming a rotational velocity $\omega$ equal to 1510 rpm, the runner diameter $D$ and the width $B$ resulting from the procedure described in the second paragraph are equal to 204 and 62 mm, respectively. The PRS casing is made of cast iron and the runner, made of stainless steel, has 40 semicircular blades [27] connected to each other by a couple of circular plates fixed to the shaft, which rotates on two bearings. There is no internal shaft. The flap is made of stainless steel and is moved by a linear electrical actuator.

Small traditional hydroelectric plants are equipped with a synchronous by-pass to stop rotation of the runner in the case of failure of the electric network. This is a pipe parallel to the runner, equipped with an automatic valve, which opens to allow the entire flow to bypass the turbine when electricity is missing. In the Fontes Episcopi PRS plant an alternative solution was selected. Between the runner shaft and the electric generator, a negative electric-brake was installed. In the case of failure of the electrical grid or an emergency, the brake is activated instantaneously to stop rotation of the runner rapidly. The total flow will continue to pass through the runner, which will have zero speed. Observe that this solution guarantees water supply even in the absence of electricity production, without installing an automatic synchronous valve.

For electricity production, an asynchronous generator 4-pole IE2 efficiency class with 55 kW power was installed. The power electronics system described in paragraph 3, with a maximum electrical power of 60 kW, was connected to the electric generator. The power electronics was oversized compared to the generator power to ensure system security. In Figure 6 the PRS turbine prototype installed inside the Fontes Episcopi building is shown.

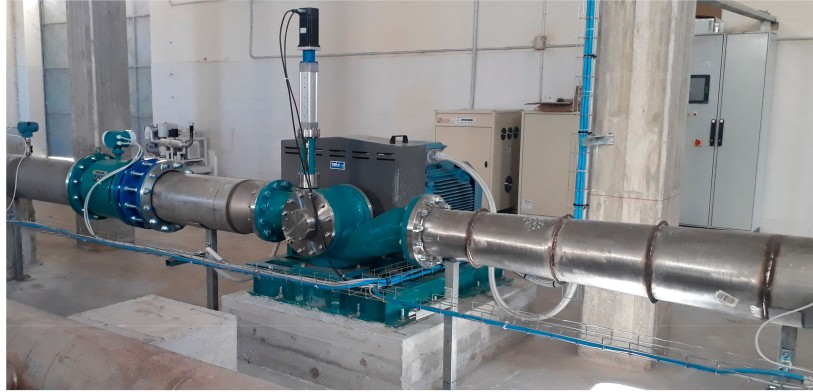

**Figure 6.** PRS turbine prototype installed in the study case.

For monitoring the hydraulic parameters, an electromagnetic flow meter and a digital pressure meter were installed upstream of the PRS prototype and a second digital pressure meter was installed downstream of the turbine to measure the net head of the turbine (Figure 7).

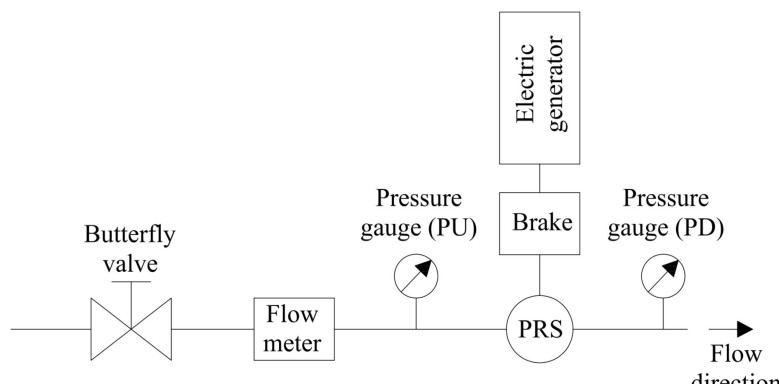

**Figure 7.** Equipment installation scheme.

The PRS components of the pilot plant are automatically regulated by a PLC installed on the electrical panel dedicated to turbine management. If the device is used in "active" mode and the flow rate $Q_{set}$ is set, the flap position is found by comparing the measure of the flow meter with its target value; if the device is used in "passive" mode, the flap position is found by comparing the pressure measured by the downstream or upstream pressure gauge with its pressure target value. In both cases, the runner rotational velocity is optimized by maximizing the electrical power $P_i$ coming out of the inverter, according to the $Q_{set}$ or $H_{set}$ values, calculated by the Equation (5):

$$P_i = \sqrt{3} \cdot V_{out,i} \cdot I_i \cdot \cos\varphi \tag{5}$$

where $V_{out,i}$ and are respectively the voltage and the current coming out of the inverter and $\cos\phi$ is the power factor.

The logic control implemented in the PLC is based on two separate controllers (SISO-controllers) running in series. The first controller searches the position of the flap for assigned hydraulic set-point. After that, for fixed position of the flap, the second controller searches the optimal rotational velocity maximizing the electric power. The flow chart of the logic control is represented by the flow chart in Figure 8.

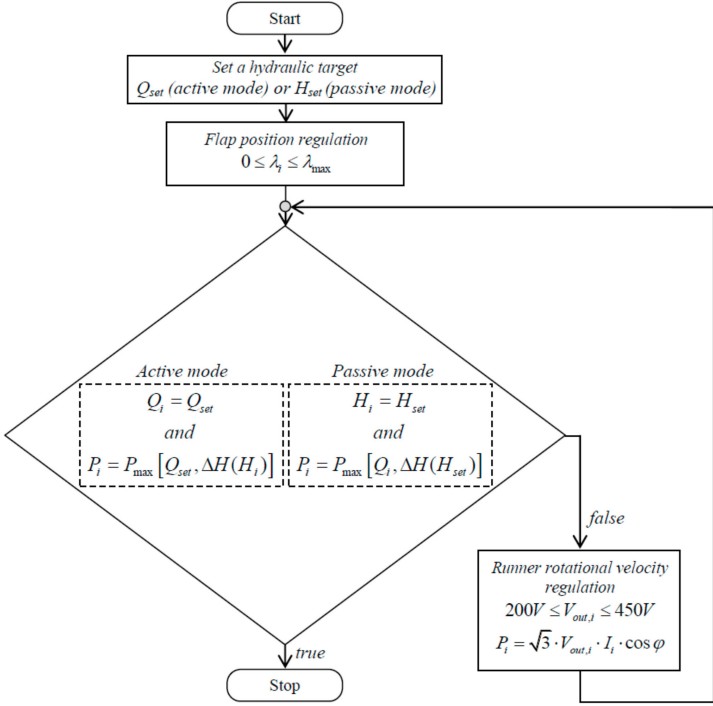

**Figure 8.** Flow chart of PRS regulation.

The hydroelectric production performance of the plant is calculated by comparing in each time the electrical output power from the inverter with the gross hydraulic power computed from the flow and pressure measurements.

## 5. PRS Turbine Application Results

During the start-up period, in order to guarantee the quality of water distribution and ensure the safety of the pipeline, the water manager needs to guarantee the following operating conditions: (1) a pressure in the range of 0.2–0.4 MPa downstream of the Fontes Episcopi building; (2) a pressure lower than 1.0 MPa on the entire supply line; (3) variable flow rate according to the given demand and in any case lower than 75 l/s. Under these operation conditions, different from the turbine design values, the PRS start-up tests were carried out.

In the following sections, the hydraulic and power variables recorded during the start-up tests on the PRS plant installed at the Fontes Episcopi building are shown. Due to the long time required by bureaucracy for connection to the Italian national electric grid and electricity trading, the electrical power produced by the plant during two days of the start-up tests was temporarily dissipated through electrical resistances.

During the start-up period, the device was set in passive mode, with the flow rate imposed by the water manager through the needle valve and shown in Figures 9 and 10. Observe that with the given flow rate, free surface conditions always occur inside the upper part of the pipeline. The pressure immediately upstream of the PRS was set according to the manager's request, given the downstream pressure curve plotted in the same figures. On the first day of testing the maximum upstream pressure was set at 0.8 MPa; on the second day of testing it was set at 1.0 MPa. The time series of the hydraulic data recorded during the testing period are all shown in Figures 9 and 10.

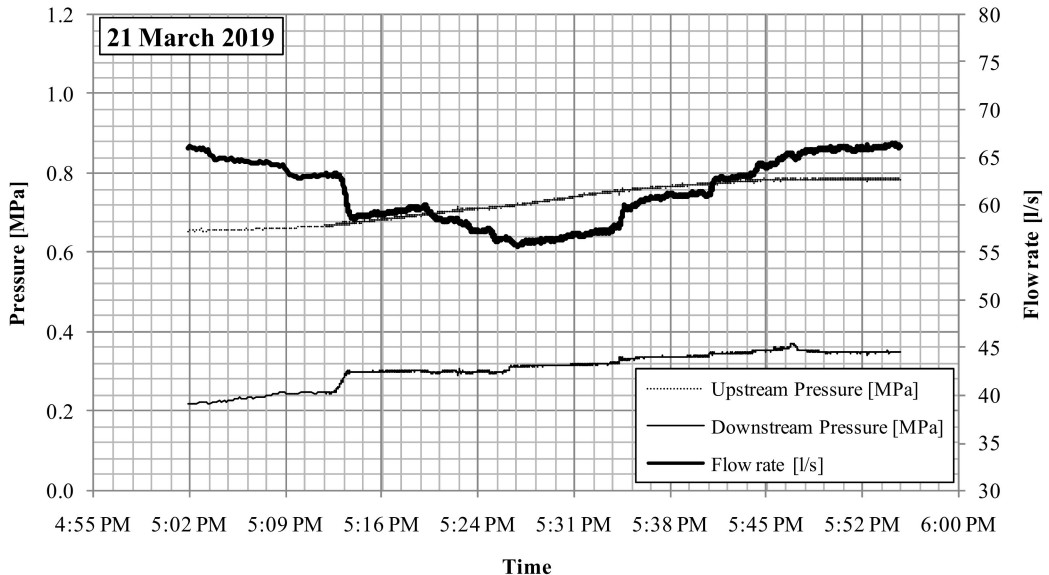

**Figure 9.** Flow rate and pressure in the manometers showed in Figure 7.

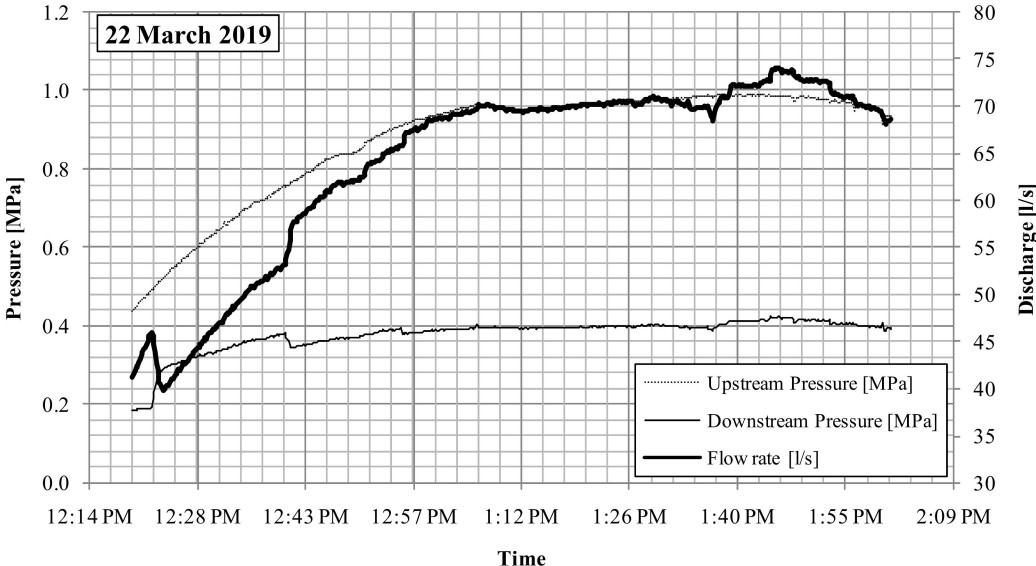

**Figure 10.** Flow rate and pressure in the manometers showed in Figure 7.

In order to evaluate the global performance of the PRS and the hydroelectric plant, voltage and current measurements were made at the input and output of the inverter, to get the electrical power along the test time. Knowledge of the generator characteristic curve made it possible to determine the efficiency of the asynchronous generator as a function of the power supplied by the generator itself. The inverter's efficiency was estimated by comparing its input and the output power. The electrical efficiencies are shown in Figure 11. The graph shows that the inverter has lower efficiency than the electric generator, but that it is constant with respect to the supplied power.

The hydraulic efficiency of the PRS was computed as the ratio between the output electric power of the generator and the available gross hydraulic power, multiplied by the total electrical efficiency. The tests carried out show an average hydraulic efficiency of 61% on the first day and 55% on the second day of operation. The hydraulic efficiency of the PRS versus time is shown in Figures 12 and 13.

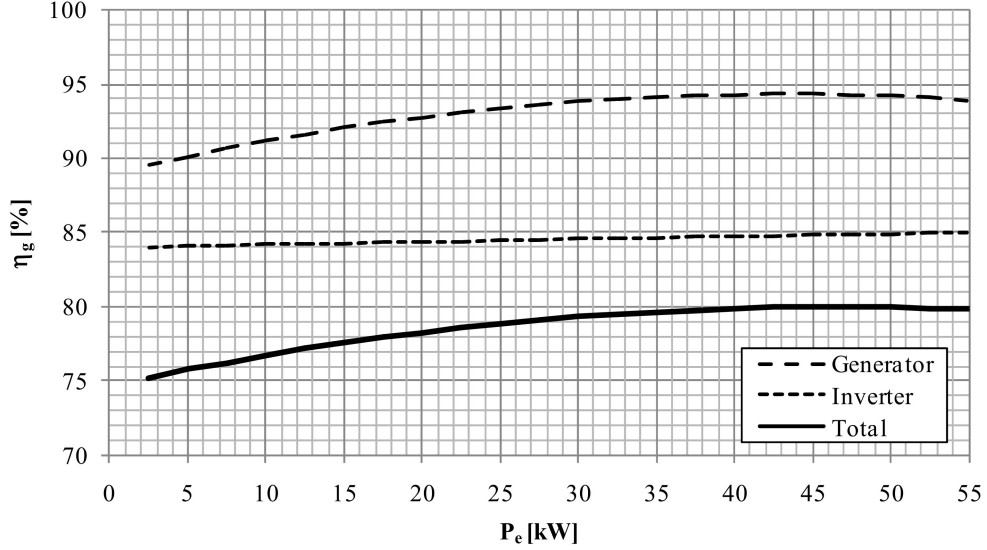

**Figure 11.** Electrical efficiencies.

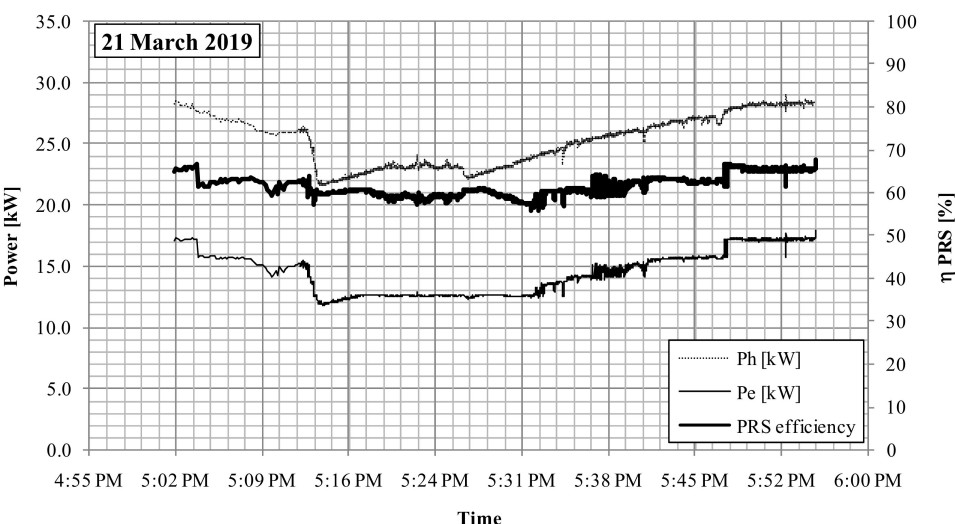

**Figure 12.** Hydraulic power, electrical power and PRS efficiency.

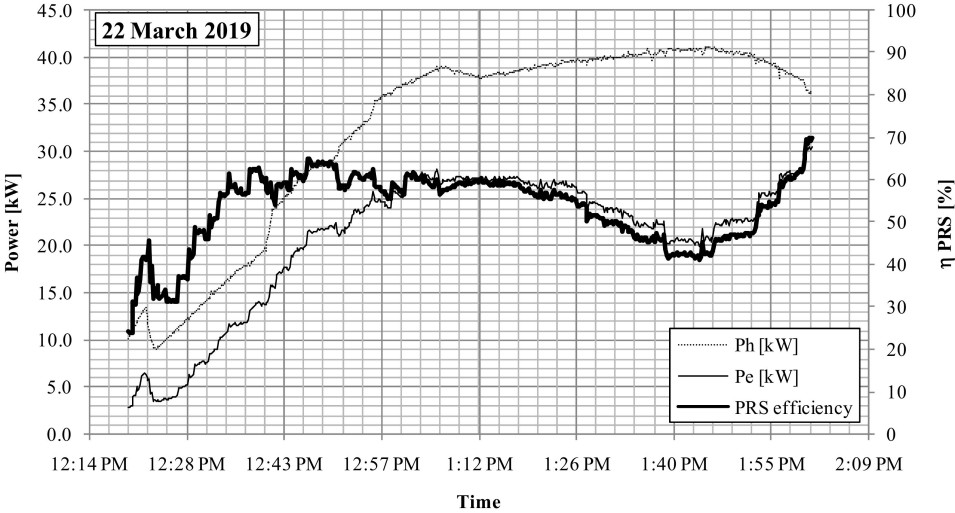

**Figure 13.** Hydraulic power, electrical power and PRS efficiency.

Some electrical disconnections of the generator were carried out during the start-up period in order to validate the effect of brake action on the water supply and on the pipeline, for different flow rate and pressure values. The tests confirmed the absence of overpressure in the pipeline generated by the instantaneous stop of the runner and validated the 30% increment of the maximum flow rate, as already numerically predicted by previous studies [24].

## 6. Conclusions

A new Banki-type turbine with positive outlet pressure, called PRS, was installed in a real water transport network for pressure regulation. The PRS is equipped with an internal flap for flow rate or pressure regulation and an inverter for the runner rotational velocity regulation. Start-up tests showed that the PRS could be efficiently used in water distribution networks for regulation of flow rate, as an alternative to needle valves, or for the regulation of the downstream/upstream head as an alternative to PRV valves. The tests also showed that the PRS is able automatically to adjust the position of its flap and optimize power production by rotational velocity regulation, according to the pressure set-point required by the water manager and the instantaneous flow rate. The simulation of the interruption of the electrical network also showed that the PRS braking system is able to quickly interrupt runner rotation, without generating overpressure on the water network. The transition of the maximum flow through the stopped runner provides a net head which is equal to the net head occurring at the optimal rotating velocity divided by 1.71, as already predicted in a previous study.

The hydraulic constraints imposed by the water manager during the start-up period did not allow use of the turbine according to the design conditions, but this is unfortunately the most common situation. In spite of that, the PRS mean efficiency, equal to 53% on the first testing day and 61% on the second testing day, coupled with a total electrical efficiency in the order of 80%, still leads to a significant amount of energy and a corresponding gain for the water manager. The cost of installing the PRS is certainly superior to the installation of a simple dissipation device, but the significant electricity production that can be obtained from the PRS guarantees a financial benefit that is significantly higher than the installation costs in the case study.

**Author Contributions:** All authors contributed to the development of this manuscript. M.S., C.A. and T.T. designed and supervised the hydraulic tests. P.A. designed the PRS turbine and supervised the mechanical components. M.F. designed electrical control systems and supervised the electrical tests.

**Funding:** The experimental plant was funded by "Pressure Management System (PMS) project, grant number no. F/050304/01-03/X32-Ministero dello Sviluppo Economico D.M. 1 Giugno 2016 Horizon 2020-PON 2014/2020".

**Acknowledgments:** We thank the BitControl srl, Layer Electronics srl and Vettorello srl companies, partners in the PMS project, for authorization of scientific use of the experimental results.

**Conflicts of Interest:** The authors declare no conflict of interest.

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
