# Peer review of "Coupled Electric and Hydraulic Control of a PRS Turbine in a Real Transport Water Network"

_water, doi:10.3390/w11061194_

Round 1

Reviewer 1 Report

The paper presents a very interesting and actual subject of energy harvesting in water distribution networks.

Only a few comments and recommendations are listed below:

- use "flow rate" instead of "discharge". Discharge is correct only when used with the meaning of "exit flow rate".

-   in line  53, "to produce hydropower" is not correct. I suggest rephrasing: "to produce energy from hydro sources".

- the references for figures should be corrected (erase Fig. and leave only Figure)

-line 79, "former" should be replaced with "first mode" or "passive mode"

- "impeller" should be replaced with "runner"

- line 167, "cited" should be replaced with "mentioned"

- line 235, "discharge variable" should be replaced with "variable flow rate"

- figs 9 and 10, PU and PD should be explained

- line 279, "ware" should be "were"

Author Response

Please, fid the replies in the attached file

Reviewer 2 Report

The paper is well written and clearly structured, and the topic is interresting as well as important. 

However, i am missing a few details on the control concept:

(1) How are the MIMO-controllers designed (decentralized, real MIMO)?

(2) Are there timeseries of the controller outputs to the flap and the rectifier/Inverter available?

(3) How is the stable operation of the asynchronous generator guaranteed, e.g., is the upper margin of the output voltage Vout variable with omega?

At least a brief sketch of the control design, in addition to the very coarse Information from Fig. 8, would be valuable, at least as the title of the paper promises some insight here.

The results, though very convincing, seem to miss a short overview of achievable efficiencies with concurrent approaches, such like PATs.

Author Response

Please, find the replies in the attached file
